# Antiproliferative and Proapoptotic Effects of Erucin, A Diet-Derived H_2_S Donor, on Human Melanoma Cells

**DOI:** 10.3390/antiox12010041

**Published:** 2022-12-26

**Authors:** Daniela Claudia Maresca, Lia Conte, Benedetta Romano, Angela Ianaro, Giuseppe Ercolano

**Affiliations:** Department of Pharmacy, School of Medicine and Surgery, University of Naples Federico II, 80131 Naples, Italy

**Keywords:** hydrogen sulfide, melanoma, erucin, nutraceuticals, cancer therapy

## Abstract

Melanoma is the most dangerous form of skin cancer and is characterized by chemotherapy resistance and recurrence despite the new promising therapeutic approaches. In the last years, erucin (ERU), the major isothiocyanate present in *Eruca sativa*, commonly known as rocket salads, has demonstrated great efficacy as an anticancer agent in different in vitro and in vivo models. More recently, the chemopreventive effects of ERU have been associated with its property of being a H_2_S donor in human pancreatic adenocarcinoma. Here, we investigated the effects of ERU in modulating proliferation and inducing human melanoma cell death by using multiple in vitro approaches. ERU significantly reduced the proliferation of different human melanoma cell lines. A flow cytometry analysis with annexin V/PI demonstrated that ERU was able to induce apoptosis and cell cycle arrest in A375 melanoma cells. The proapoptotic effect of ERU was associated with the modulation of the epithelial-to-mesenchymal transition (EMT)-related cadherins and transcription factors. Moreover, ERU thwarted the migration, invasiveness and clonogenic abilities of A375 melanoma cells. These effects were associated with melanogenesis impairment and mitochondrial fitness modulation. Therefore, we demonstrated that ERU plays an important role in inhibiting the progression of melanoma and could represent a novel add-on therapy for the treatment of human melanoma.

## 1. Introduction

Recognized as the most dangerous and fatal form of skin cancers, the incidence of melanoma is swiftly increasing in the entire world, and it mostly affects people with a median age of 57 years and with a fair phototype [1,2]. Melanoma is characterized by the dysfunctional proliferation of melanocytes, cells located in the basal layer of the epidermis which produce the melanin pigment, which are essential for skin protection from solar radiation. Excessive exposure to ultraviolet rays following sunbathing or tanning represents the major risk factor of melanoma [3]. Indeed, UVA and UVB rays cause DNA damage in skin cells which leads to specific mutations that, when accumulated or are in excess, may not be sheltered by DNA repair systems [4]. This triggers the development of genetic alterations, such as the tumor suppressor gene TP53 mutation, the CDK inhibitor gene CDKN2A mutation or the activating BRAF^V600E^ mutation [5,6]. The latter is not typically a UV-induced mutation although a potential sunlight-mediated origin has been described [7,8]. The BRAF^V600E^ mutation is harbored by more than 50% of melanoma patients, inducing uncontrolled cancer cell proliferation, migration and metastasis development [9]. In fact, BRAF inhibitors as well as the most recently approved anti-PDL1 and anti-CTLA-4 monoclonal antibodies represent the first line treatment for melanoma cancer patients. Nevertheless, an important percentage of patients become nonresponsive to these treatments, developing tumor resistance and exacerbation [10]. Therefore, there is an increasing interest in finding and characterizing new promising agents that can be proposed as novel therapeutic approaches for the treatment of melanoma in order to overcome tumor resistance and recurrence. In this context, the use of diet-derived phytochemicals has been proposed as an adjuvant therapeutic approach for cancer treatment. Different diet-derived compounds, such as resveratrol, curcumin, apigenin and capsaicin demonstrated potent bioactivities, showing anti-inflammatory, antibiotic and antitumoral effects [11]. In the last few years, plant and food research has been focused on cruciferous-vegetable-derived compounds. In particular, cruciferous vegetables, which include rocket salads, broccoli and cabbage, are rich in isothiocyanates, which are the hydrolysis products of glucosinolate. Importantly, isothiocyanates have been proposed as promising chemopreventive agents for modulating cancer development and progression as demonstrated by multiple in vitro and in vivo approaches [12,13]. 4-(methylthio) butyl isothiocyanate, commonly referred to as erucin (ERU), is the major component derived from rocket salad leaves and was demonstrated to affect the proliferation of different cancer cell lines through different mechanisms of action, such as apoptosis, autophagy, cell cycle arrest and antioxidant effects [14,15,16,17]. Importantly, Citi et al. recently demonstrated that the antiproliferative and proapoptotic effects of ERU in human pancreatic carcinoma cells are ascribed to the ability of ERU to release hydrogen sulfide (H_2_S) [18,19]. Likewise, our own group, and others, showed that the H_2_S pathway participates in melanoma progression and demonstrated that exogenous H_2_S, by means of H_2_S-releasing molecules, represents a promising therapeutic approach for the management of metastatic melanoma [20,21,22,23,24,25]. In this study, we characterized the therapeutic potential of ERU in human melanoma for the first time by evaluating its ability to modulate the proliferation, migration, ROS production and mitochondrial activity of human melanoma cells in vitro.

## 2. Materials and Methods

### 2.1. Reagents and Cell Culture

The human melanoma cells lines A375 were bought from Sigma–Aldrich (Milan, Italy). WM1862, WM983A and WM983B cell lines were purchased from Rockland (Limerick, Ireland). Human keratinocytes (HaCaT) were purchased from Lonza (Basel, Switzerland)). All cell lines were cultured in RPMI 1640 medium with GlutaMAXTM and were supplemented with 10% heat-inactivated fetal calf serum, 2 mmol/L L-glutamine, 100 U/mL penicillin, 100 μg/mL streptomycin and 10 mM HEPES buffer (all from Gibco; New York, NY, USA). Cells were grown at 37 °C in a humidified incubator under 5% CO_2_. All cell lines used in this study were characterized by the cell bank where they were purchased. ERU was purchased from Cayman Chemicals (Michigan, CA, USA).

### 2.2. Proliferation Assay

Cell proliferation was measured with the 3-[4,5-dimethyltiazol2yl]-2,5 diphenyl tetrazolium bromide (MTT) assay. Different human melanoma cell lines (A375, WM1862, WM983A and WM983B) and normal keratinocytes (HaCaT) were seeded on 96-well plates (3 × 10^3^/well) and were treated with different concentrations of ERU (5, 10, 20, 40 and 80 μM) for 48 h before adding 25 μL of MTT (Sigma, Milan, Italy) (5 mg/mL in saline). Thereafter, cells were incubated for 3 h at 37 °C and were then lysed in order to solubilize the dark blue crystals with a solution containing 50% (*vol*/*vol*) N,Ndimethylformamide and 20% (*wt*/*vol*) sodium dodecylsulfate with an adjusted pH of 4.5. The OD of each well was obtained by measuring the absorbance at 620 nm using Multiskan GO microplate reader (Thermo Fisher Scientific, Waltham, MA, USA).

### 2.3. Flow Cytometry Analysis

To assess cell proliferation, A375 cells were incubated with 5 μM carboxyfluorescein succinimidyl ester (CFSE, Thermo Fisher Scientific, Waltham, MA, USA) for 20 min and were either directly analyzed or grown for 48 h in the presence of 30 µM ERU before analyzing fluorescence intensity through flow cytometry.

Ki67 expression was evaluated through intracellular staining performed after fixation and permeabilization with Intracellular Fixation & Permeabilization Kit (eBioscience, Thermo Fisher Scientific Waltham, MA, USA) using APC antihuman Ki67 antibodies (REA183, Miltenyi, 2:50).

For live vs. dead status, 48 h ERU-treated A375 cells (30 µM) were labeled with the Zombie Green Fixable Viability Kit (BioLegend, San Diego, CA, USA) and washed as instructed by the manufacturer’s instructions.

Apoptosis assay was performed by using the Annexin V-FITC Kit (BD Pharmingen, San Diego, CA, USA) according to the manufacturer’s instructions. Briefly, A375 cells were seeded on a 6-well plate (3 × 10^5^ cells/well) and were allowed to attach overnight. The day after, cells were treated with 30 μM ERU and were incubated for 48 h. Subsequently, cells were collected and stained for 10 min with FITC-conjugated annexin V. Then, the samples were washed and stained with propidium iodide before flow cytometry analysis.

Cell cycle analysis was performed using Cell Cycle Assay Solution Deep Red Kit (Dojindo, Kumamoto, Japan). The 24 h ERU-treated A375 cells (30 µM) were incubated with Cell Cycle Assay Solution (5 µL) in PBS for 15 min at 37 °C before flow cytometry analysis.

For E-CAD and N-CAD expression, 24 h ERU-treated A375 cells (30 µM) were incubated with the following antibodies: APC-Cy7 antihuman CD324 (E-CAD) (67A4, Biolegend, 1:50) and Alexa-Fluor 700 antihuman CD325 (N-CAD) (8C11, Biolegend, 2:50). Cells were stained with FACS buffer for 20 min at room temperature.

For ROS production and mitochondrial activity, 48 h ERU-treated A375 cells (1 µM) were stained with H2DCFDHA (D399 Thermo Fisher Waltham, MA, USA), MitoTracker Green (M7514 Thermo Fisher Waltham, MA, USA) and MitoTracker Deep Red (M22426, Thermo Fisher Waltham, MA, USA) as previously described [26].

Samples were acquired on BriCyte E6 flow cytometer (Mindray Medical Italy S.r.l., Milan, Italy), and data were analyzed using FlowJo software (TreeStar V.10; Carrboro, NC, USA). All the histograms were edited with modal option.

### 2.4. Caspase 3/9 Activity Assay

Activation of Caspase 3 and 9 in ERU-treated A375 cells (30 µM for 48 h) were determined with Caspase 3 and 9 Activity Colorimetric Assay Kits according to the manufacturer’s instructions (Houston, Texas, 77079, USA).

### 2.5. RNA Extraction and Quantitative Real-Time PCR (qPCR)

Total RNA was isolated from A375 melanoma cells treated or not treated with 1 µM or 30 µM ERU for 6 h using the TRIZOL reagent (Invitrogen, Thermo Fisher Scientific Waltham, MA, USA ) as previously described. RNA was quantified with Nanodrop and considered DNA- and protein-free if the ratio of readings at 260/280 nm was ≥1.7. Isolated mRNA was reverse-transcribed by iScript Reverse Transcription Supermix for RT-qPCR (Bio-Rad, Milan, Italy). qPCR was carried out in the Bio-Rad CFX384 real-time PCR detection system (Bio-Rad, Milan, Italy) with the following primers:BCL-2 (Gene ID: 596)5′-GGTGGGGTCATGTGTGTGG-3′;5′-CGGTTCAGGTACTCAGTCATCC-3′XIAP (Gene ID: 331)5′-TATCAGACACCATATACCCGAGG-3′;5′-TGGGGTTAGGTGAGCATAGTC-3′CCNB1(Gene ID: 891)5′-GACCTGTGTCAGGCTTTCTCTG-3′;5′- GGTATTTTGGTCTGACTGCTTGC-3′CDK1(Gene ID: 983)5′-GGAAACCAGGAAGCCTAGCATC-3′;5′-GGATGATTCAGTGCCATTTTGCC-3′CDC25C (Gene ID: 995)5′-TCTACGGAACTCTTCTCATCCAC-3′;5′-TCCAGGA CAGGTTTAACATTTT-3′SNAIL (Gene ID: 6615)5′-ACTGCAACAAGGAATACCTCAG-3′;5′-GCACTGGTACTTCTT GACATCTG-3′SLUg (Gene ID: 6591)5′-CGAACTGGACACACATACAGTG-3′;5′-CTGAGGATCTCTGGTTGTGGT-3′ZEB-1 (Gene ID: 6935)5′-TTACACCTTTGCATACAGAACCC-3′;5′-TTTACGAT TACACCCAGACTGC-3′TWIST (Gene ID: 7291)5′-GTCCGCAGTCTTACGAGGAG-3′;5′-GCTTGAGGGTCTGAATCTTGCT-3′GCLC (Gene ID: 2729)5′GTTGGGGTTTGTCCTCTCCC-3′;5′-GGGGTGACGAGGTGGAGTA-3′GCLM (Gene ID: 2730)5′-AGGAGCTTCGGGACTGTATCC-3′;5′-GGGACATGGTGCATTCCAAAA-3′HMOX-1(Gene ID: 3162)5′-GCCGTGTAGATATGGTACAAGGA-3′;5′-AAGCCGAGAATGCTGAGTTCA-3′MITF (Gene ID:4286)5′-TGGTTTTCCCACGAGCTATTTT-3′;5′-GCACAGAG TCAATTTCCTGGT-3′TYR (Gene ID: 7299)5′-GCAAAGCATACCATCAGCTCA-3′;5′-GCAGTGCATCCATTGACACAT-3′

The housekeeping gene ribosomal protein S16 (RPS16) was used as an internal control to normalize the Ct values using the 2^−ΔCt^ formula.

### 2.6. Migration Assay

A375 cells were plated in 12-well plates (2 × 10^5^ cells/well) and were allowed to grow at confluence. Subsequently, a wound was created in the monolayer using a 200 μL pipette tip, and microscope photos were taken to mark the initial condition (time 0). Cells were treated with 1 µM ERU, and, after 24 and 48 h scratches were photographed. ImageJ’s MRI Wound Healing Tool (MRI Redmine) was used to calculate the area of the cell-free gap.

### 2.7. Clonogenic Assay

A375 cells were plated in a 6-well plate (1 × 10^3^ cells/well) and were treated with 1 µM ERU for 48 h. Next, fresh medium without ERU was changed every 2 days. After 14 days, colonies were formed, and the cells were washed with PBS, fixed with 4% paraformaldehyde and stained with 0.5% crystal violet. Colonies were manually counted, and images were acquired using a digital camera.

### 2.8. Invasion Assay

Boyden chambers with polycarbonate filters with a nominal pore size of 8 μm (Millipore, USA) were coated on the upper side with Matrigel (Becton Dickinson Labware, USA). The chambers were placed in a 24-well plate, and A375 cells (2.5 × 10^5^ cells/mL) were plated in the upper chamber in the presence or absence of ERU (1 μM) in serum-free RPMI. At the end of the 16 h of incubation, the medium was removed, and the filters were fixed with 4% formaldehyde for 2 min, and, subsequently, the cells were permeabilized with 100% methanol for 20 min. The methanol was removed, and the chambers were stained with Giemsa for 15 min and then washed with PBS. The filters were removed, and the nonmigrating cells on the top of the filter were peeled off with the use of a cotton swab. Then, the filters were placed on a slide and were examined under a microscope. Cell invasion was determined by counting the number of cells stained on each filter in at least 4–5 randomly selected fields. The resulting data were presented as the average of the invaded cells ± SEM/microscopic field of three independent experiments.

### 2.9. Measurement of Melanin Content

A375 cells (3 × 10^5^/well) were plated in a 6-well plate and treated with 1 µM ERU for 72 h. Thereafter, cell pellet was dissolved in NaOH 1N, and it was incubated for 90 min at 37 °C then centrifuged for 10 min at 10,000× *g*. The optical density (OD) of supernatant was measured at 450 nm using Multiskan GO microplate reader (Thermo Fisher Scientific, Waltham, MA, USA).

### 2.10. Statistical Analysis

Statistical analysis was performed using GraphPad Prism software version 9 (San Diego, CA, USA). For comparison of two groups, a t test was used, and, for comparison of multiple groups ANOVA test was used. The data were shown as mean ± SEM. A *p* value < 0.05 was considered statistically significant and was labeled with *; *p* values < 0.01, 0.001 or 0.0001 were labeled with **, *** or ****, respectively.

## 3. Results

### 3.1. ERU Affected the Proliferation Rate of Human Melanoma Cell Lines

First, we assessed the antiproliferative effects of ERU on different human melanoma cell lines that featured the BRAF^V600E^ mutation. WM1862, WM983A, WM983B and A375 human melanoma cells were treated with an increasing concentration of ERU (0, 5, 10, 20, 40 and 80 µM) for 48 h prior to the evaluation of cell proliferation with an MTT analysis. As shown in Figure 1A, ERU significantly reduced the proliferation of all the melanoma cell lines that were tested. In particular, 80 µM ERU reduced the cell viability by about 50% in the WM983A and WM1862 cells (*p* < 0.0001 compared with the untreated cells) and by more than 70% in the A375 and WM983B cells (*p* < 0.0001 compared with the untreated cells). Conversely, a reduction of about 20% was observed for our negative control, which was represented by human keratinocytes (HaCaT) (*p* < 0.01 compared with the untreated cells). In fact, an IC50 analysis (Figure 1B) showed a value higher than 100 µM for the HaCaT cells, whilst, for the melanoma cancer cells, it was between 30 and 60 µM, suggesting that the antiproliferative effect of ERU at lower concentrations was specific to the cancer cell. Given that the MTT assay measures the cytotoxic effect by assessing the mitochondrial dehydrogenase activity, we decided to directly evaluate the antiproliferative effect of ERU by performing the carboxyfluorescein succinimidyl ester (CFSE) assay on the A375 melanoma cell lines that showed the lower IC50 values and displayed a more aggressive phenotype compared to the other melanoma cell lines that were tested [27]. As shown in Figure 1C,D, the CFSE fluorescence intensity was strongly reduced in the control cells at 48 h compared to the control cells at t0, confirming the high proliferation rate of the A375 melanoma cell line. Importantly, the CFSE fluorescence intensity was significantly higher in the ERU-treated cells compared to the control cells at 48 h, indicating that ERU reduced the proliferation rate of the A375 melanoma cells. To corroborate our findings, we evaluated the expression of Ki67, one of the key markers involved in cancer cell proliferation [28]. As shown in Figure 1E,F, the treatment with 30 µM ERU significantly reduced the expression levels of Ki67 as observed through the flow cytometry analysis. These results demonstrated that ERU at lower concentrations inhibited the proliferation rate of the human melanoma cells without affecting the proliferation rate of the healthy control cells.

### 3.2. ERU Induced Apoptosis and the Cell Cycle Arrest of Human Melanoma Cells

Next, we decided to evaluate whether the antiproliferative effect of ERU was due to the induction of cell death by apoptosis and/or necrosis. First, we assessed the live vs. dead status of the ERU-treated A375 melanoma cells using the fluorescent dye Zombie Green. As shown in Figure 2A,B, the ERU treatment for 48 h significantly induced cell death in the A375 melanoma cells compared to the control. Next, to define whether the A375 dead status was associated with apoptosis and/or necrosis, we performed an annexin V and PI double staining analysis (Figure 2C). A FACS analysis showed that ERU significantly induced the apoptosis of the A375 cells, confirming that cell death was mediated by apoptosis (Figure 2D). To support this finding, we monitored the activation of caspase 9 and 3, which are the key players in the upstream and downstream regulation of apoptotic signal transduction [29]. As expected, the 48 h treatment with 30 µM ERU significantly induced the activation of both caspase 9 and 3 (Figure 2E). In addition, we also assessed the expression of two antiapoptotic genes, the X-chromosome-linked inhibitor of the apoptosis protein (XIAP) and B-cell lymphoma gene 2 (Bcl-2). A qPCR analysis showed that ERU markedly decreased the expression of both antiapoptotic genes (Figure 2F) in the A375 human melanoma cells. Furthermore, we also performed a cell cycle assay to evaluate the cell cycle distribution of the A375 cells after a 24 h treatment with ERU (Figure 2G). The ERU treatment significantly increased the percentage of the A375 cells in the G2/M phase and reduced the percentage of the A375 cells in the G1 and S phases compared to the control (Figure 2H). To confirm this result, we also analyzed the expression of CCNB1, CDK1 and CDC25C, the most important cell cycle regulatory proteins involved in the regulation of G2/M progression [30]. As shown in Figure 2I, the mRNA levels of CCNB1, CDK1 and CDC25C were significantly decreased in the ERU-treated A375 cells compared to the control. These results demonstrated that ERU exerted a proapoptotic effect in the melanoma cells and induced their cell cycle arrest in the G2/M phase.

### 3.3. ERU Modulated the Expression of Cadherins in Human Melanoma Cells

Apoptosis and the cell cycle are complex mechanisms that are finely tuned by different pathways. Among these, cadherins are key players involved in the phenomenon of the epithelial-to-mesenchymal transition (EMT) that favor cancer cell proliferation and invasion. In particular, the loss of E-cadherin (E-CAD) from cancer cells is associated with apoptosis inhibition, whilst the increase of N-cadherin (N-CAD) promotes cancer cell growth. Moreover, it has been also demonstrated that both N-CAD and E-CAD are associated with apoptosis given their ability to increase proapoptotic genes [31] or to interact with death receptors [32], respectively. Thus, we evaluated whether the expression of both E-CAD and N-CAD was modulated in the ERU-treated A375 melanoma cells. As shown in Figure 3A–C, ERU promoted the expression of the epithelial protein E-CAD and reduced the expression of the mesenchymal protein N-CAD after the 24 h treatment of A375 melanoma cells. Moreover, we analyzed the expression of transcription factors associated with the EMT and apoptosis resistance (e.g., SNAIL, SLUG, ZEB1 and TWIST) [33]. A qPCR analysis demonstrated that ERU significantly reduced the expression of all the transcription factors tested in the A375 melanoma cells (Figure 3D). These data further demonstrated that the proapoptotic effect of ERU was associated with the modulation of the EMT-related cadherins.

### 3.4. ERU at Low Concentrations Impaired Melanoma Cell Migration and Invasiveness

In line with the findings about cadherins, we decided to evaluate whether ERU at low concentrations below the IC50 value was able to affect the migration and invasion of the A375 melanoma cells. Thus, we decided to use the concentration of 1 µM which showed no cytotoxic effect on the A375 melanoma cells. First, we performed a migration assay on the A375 melanoma cells treated with 1 µM ERU. As shown in Figure 4A,B, ERU significantly reduced the migration of the A375 melanoma cells at both 24 and 48 h. Likewise, the colony formation assay confirmed that ERU reduced the number of the A375 colonies compared to the control (Figure 4C,D). In addition, the invasion assay demonstrated that ERU significantly reduced the invasiveness of the A375 melanoma cells (Figure 4E,F). Melanin production by cancer cells was reported to promote melanoma progression and metastasis development by affecting the different molecular mechanisms including the EMT [34]. Therefore, we hypothesized that ERU could modulate melanin production in the A375 cells. To address this point, we evaluated the melanin content in the ERU-treated A375 cells. As shown in Figure 4G, we found that ERU significantly reduced the melanin content in the A375 melanoma cells. To corroborate this finding, we evaluated the mRNA expression level of the microphthalmia-associated transcription factor (MITF) and the tyrosinase enzyme (TYR), the two most important genes involved in melanin synthesis and in melanoma development [35], with a qPCR analysis. As expected, 1 µM ERU significantly reduced the expression levels of both the MITF and TYR (Figure 4H). Taken together, our data suggested that ERU thwarted the migratory and invasive capacity of the melanoma cells by modulating their melanin production.

### 3.5. ERU Inhibited ROS Production in Melanoma Cells by Limiting Their Mitochondrial Function

It has been described that the presence of melanin inside melanoma cells triggers the production of important levels of reactive oxygen/nitrogen species (ROS/RNS), promoting melanoma progression [36]. To further dissect the mechanism underlying the antimigratory effects on the A375 cells, we evaluated the ability of ERU to modulate ROS production in the melanoma cells. Interestingly, the treatment for 48 h with ERU (1 µM) significantly suppressed ROS formation as demonstrated by the reduced fluorescence intensity of the DCF probe (Figure 5A). Next, we measured the mitochondrial mass and membrane potential of the ERU-treated A375 cells given the key role of the mitochondria in ROS production [37]. As shown in Figure 5B,C, both the Mitotracker Green and Deep Red dyes’ uptakes were significantly decreased in the ERU-treated A375 cells, suggesting a reduced mitochondrial mass and mitochondrial membrane potential. To corroborate our data, we assessed the effect of ERU on the expression of different antioxidant enzymes such as GCLC and GCLM, the catalytic and modulatory subunits involved in the synthesis of glutathione, respectively, as well as the heme oxygenase-1 enzyme (HMOX-1). As expected, a qPCR analysis showed a significant increase in GCLC, GCLM and HMOX-1 after treatment of the A375 cells with 1uM ERU (Figure 5D). These findings indicated that the antimigratory effects of ERU correlated with a decline in the mitochondrial function, which, in turn, impaired ROS production and cellular fitness in the melanoma cancer cells.

## 4. Discussion

Natural products are emerging as promising tools in cancer therapy given their multitarget activity and their ability to modulate the tumor microenvironment. In particular, diet-derived compounds, such as coffee; tea; pomegranate; extra virgin olive oil; and brassicaceae vegetables, which include broccoli, brussels sprouts and rocket salads, have been widely demonstrated to prevent cancer development [11,38]. In fact, it has been reported that melanoma and nonmelanoma skin cancers’ low incidence in Mediterranean populations might be associated with the intake of the vegetables, fish and fruit that constitute the traditional Mediterranean diet [39,40]. This is mainly due to the presence of different dietary antioxidant compounds, such as carotenoids, vitamins, polyphenols and isothiocyanates. Importantly, different data also suggested that these compounds may improve the efficacy of classic chemotherapeutics by exerting a synergistic effect on the one hand and by restraining chemoresistance on the other hand [41,42]. Particularly, the anticancer effects of isothiocyanates have been ascribed to their ability to release H_2_S [19,43]. In fact, H_2_S-releasing agents have been proposed as a promising therapeutic approach for the treatment of different types of cancer [44,45,46,47]. Likewise, ERU, the major isothiocyanate present in rocked salads with H_2_S-releasing properties [48,49,50,51,52,53], demonstrated its anticancer activity in different tumor cell lines in vitro and in tumor-bearing mice in vivo [18,54,55,56]. In this study, we characterized the anticancer properties of ERU in human melanoma by using multiple in vitro approaches. We observed that ERU inhibited the proliferation of the different human melanoma cell lines in a time-dependent manner as has also been reported in pancreatic cancer cells [18]. Moreover, similarly to other diet-derived compounds [57,58,59], ERU modulated the expression of the proliferation marker Ki67 that has been proposed as a prognostic biomarker in cutaneous melanoma [60]. Our own group, and others, demonstrated that both natural and synthetic H_2_S donors induce apoptosis in melanoma cells [20,21,22,24,25]. Likewise, we demonstrated that the antiproliferative effect of ERU was coupled with its ability to induce apoptosis in the A375 melanoma cells. In fact, we observed the activation of both caspase 3 and 9 and the downregulation of the proapoptotic genes BCL-2 and XIAP after 48 h of treatment with ERU. Moreover, ERU was able to induce cell cycle arrest in the G2/M phase as previously demonstrated in pancreatic and breast cancer cells [14,18]. E-CAD and N-CAD represented the two major proteins involved in the EMT phenomenon and was reported to orchestrate apoptosis [32,61]. ERU significantly increased the expression of the epithelial protein E-CAD, whilst it reduced the expression of the mesenchymal protein N-CAD. Melanoma progression and metastasis development were associated with the ability of melanoma cells to acquire aggressive properties, such as motility and invasion [62].

In fact, multiple diet-derived compounds were demonstrated to prevent tumor progression by modulating these parameters in different cancer cells [63,64,65,66]. In our study, we found that ERU at low concentrations below the IC50 value (1 µM) significantly reduced the migration, invasion and clonogenic potential of the A375 melanoma cells, which were parameters that reflected their ability to generate metastases in vivo. Emerging evidence demonstrated that the melanin secreted from the melanoma cells supported their progression and metastasis development, suppressing the immune response and promoting tumor angiogenesis [34,67]. Our results showed that ERU exerted an essential role in modulating the melanogenesis in the melanoma cells by inhibiting the melanin content and suppressing the expression of the MITF and TYR, which are the key factors that promote melanin synthesis [35,68]. Melanoma-cell-produced melanin also contributes to oxidative stress, which in turn promotes melanoma initiation and progression [69,70]. In our study, ERU reduced intracellular ROS generation in the A375 cells by modulating their mitochondrial activity. Moreover, ERU increased the expression of antioxidant target genes, such as GCLC, GCLM and HMOX-1, suggesting that the modulation of ROS production and the impairment of mitochondrial activity in the melanoma cells were among the contributing factors, which supported the antitumor activity of ERU.

## 5. Conclusions

This work widely characterized the anticancer properties of ERU in human melanoma in vitro. In fact, we demonstrated that, in the human melanoma cells, ERU (i) inhibited cell proliferation, (ii) induced apoptosis and cell cycle arrest and (iii) reduced the expression of cadherins and their related transcription factors. Moreover, a low concentration of ERU thwarted cell migration and invasion. This effect was associated with reduced levels of melanin and melanogenesis-associated genes. This is an important feature in melanoma progression since melanin production has been associated with the EMT and oxidative stress. Thus, ERU could represent a new promising diet-derived compound with anticancer properties, which are ascribed to its ability to release H_2_S. However, translational in vivo studies are required to gain further insight into the antitumoral effects of ERU using murine models of cutaneous and metastatic melanoma.

## Figures and Tables

**Figure 1 antioxidants-12-00041-f001:**
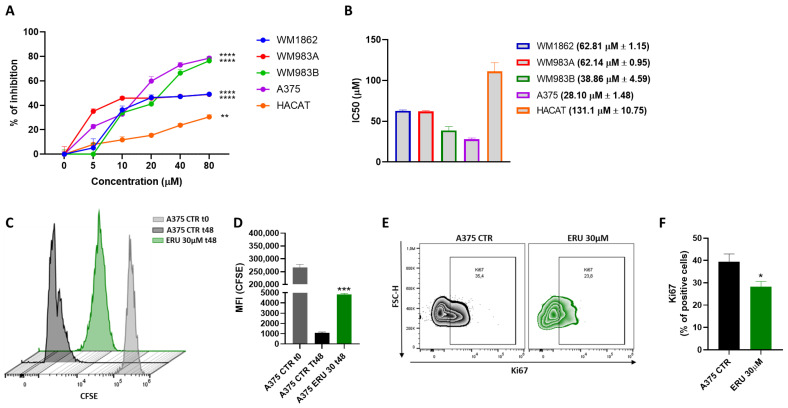
ERU affected the proliferation rate of human melanoma cell lines. (**A**) Antiproliferative effect of ERU (0–80 μM) was assessed with MTT assay in A375, WM1862, WM983A and WM983B melanoma cells and in normal human keratinocytes (HaCaT) at 48 h. (**B**) IC50 values for ERU-treated A375, WM1862, WM983A and WM983B melanoma cells and for HaCaT cells. (**C**) Representative example of flow cytometry analysis of CFSE staining in A375 cells after staining (grey histogram) and after 48 h of treatment (green histogram) or no treatment (black histogram) with 30 μM ERU. (**D**) CFSE quantification in terms of mean fluorescence intensity (MFI). (**E**) Representative example of flow cytometry analysis of A375-derived Ki67 upon treatment (green dot plot) or no treatment (black dot plot) with 30 μM ERU. (**F**) Frequency of Ki67 in A375 cells after treatment (green bar) or no treatment (black bar) with 30 μM ERU. Data were shown as mean ± SEM of at least three independent experiments (* *p* < 0.05, ** *p* < 0.01, *** *p* < 0.001 and **** *p* < 0.0001 vs. A375 CTRL).

**Figure 2 antioxidants-12-00041-f002:**
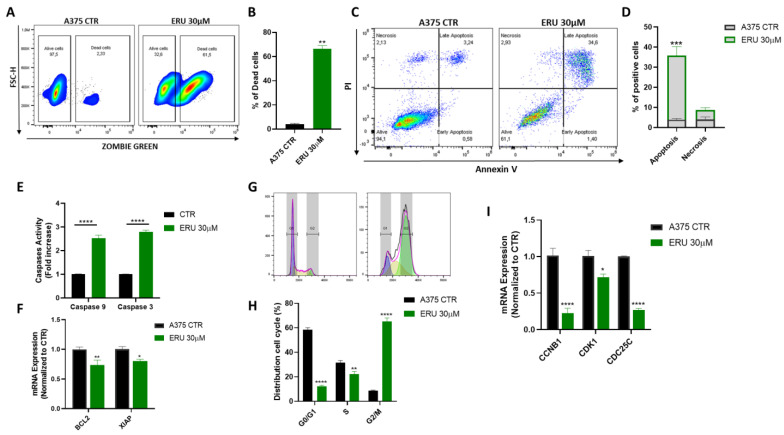
ERU induced apoptosis and cell cycle arrest of human melanoma cells. (**A**) Representative example of flow cytometry analysis of Zombie Green staining in A375 cells upon 48 h of treatment or no treatment with 30 μM ERU. (**B**) Frequency of dead cells after treatment (green bar) or no treatment (black bar) for 48 h with 30 μM ERU. (**C**) Representative example of annexin V/propidium iodide (PI) staining after 48 h of treatment or no treatment with 30 μM ERU. (**D**) Frequency of apoptotic cells after treatment (green bar) or no treatment (black bar) for 48 h with 30 μM ERU. (**E**) Activation of caspase 9 and 3 in A375 cells upon 48 h of treatment (green bar) or no treatment (black bar) with 30 μM ERU. (**F**) Expression of BCL2 and XIAP assessed with qPCR in A375 cells upon 6 h of treatment (green bar) or no treatment (black bar) with 30 μM ERU. (**G**) Representative example of cell cycle distribution in A375 cells upon 24 h of treatment or no treatment with 30 ERU. (**H**) Frequency of A375 cells in G0/G1, S and G2/M cell cycle distributions after treatment (green bar) or no treatment (black bar) for 24 h with 30 μM ERU. (**I**) Expression of CCNB1, CDK1 and CDC25C assessed with qPCR in A375 cells upon 6 h of treatment (green bar) or no treatment (black bar) with 30 μM ERU. Data were shown as mean ± SEM of at least three independent experiments (* *p* < 0.05, ** *p* < 0.01, *** *p* < 0.001 and **** *p* < 0.0001 vs. A375 CTRL).

**Figure 3 antioxidants-12-00041-f003:**
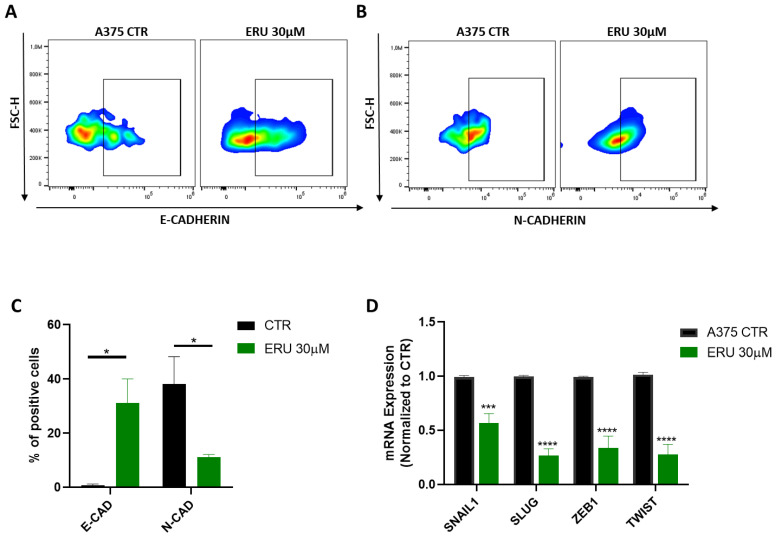
ERU modulated the expression of cadherins in human melanoma cells. (**A**,**B**) Representative example of flow cytometry analysis of A375-derived E-CAD and N-CAD upon treatment (green dot plot) or no treatment (black dot plot) with 30 μM ERU. (**C**) Frequency of Ki67 in A375 cells after treatment (green bar) or no treatment (black bar) with 30 μM ERU. (**D**) Expression of SNAIL1, SLUG, ZEB-1 and TWIST assessed with qPCR in A375 cells upon 6 h of treatment (green bar) or no treatment (black bar) with 30 μM ERU. Data were shown as mean ± SEM of at least three independent experiments (* *p* < 0.05, *** *p* < 0.001 and **** *p* < 0.0001 vs. A375 CTRL).

**Figure 4 antioxidants-12-00041-f004:**
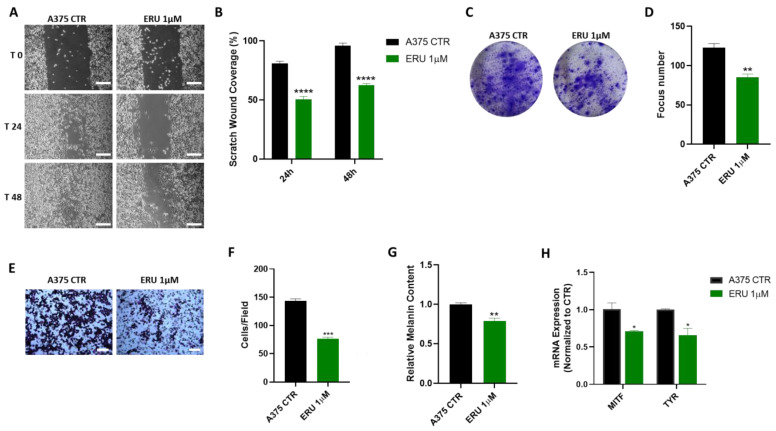
ERU at low concentrations impaired melanoma cell migration and invasiveness. (**A**) Representative example of wound healing assay of A375 cells after incubation with 1 μM ERU for 24 and 48 h (scale bar: 250 μm). (**B**) Quantification of the healed wound area at 24 and 48 h. (**C**,**D**) Representative example (**C**) and quantification (**D**) of clonogenic assays of A375 cells after incubation with 1 μM ERU. (**E**) Representative example of invasion assay of A375 cells after incubation with 1 μM ERU (scale bar: 200 μm). (**F**) Average number of invasive cells per field. (**G**) Melanin content in A375 melanoma cells upon treatment (green dot plot) or no treatment (black dot plot) with 1 μM ERU. (**H**) Expression of MITF and TYR assessed with qPCR in A375 cells upon 6 h of treatment (green bar) or no treatment (black bar) with 1 μM ERU. Data were shown as mean ± SEM of at least three independent experiments (* *p* < 0.05, ** *p* < 0.01, *** *p* < 0.001, and **** *p* < 0.0001 vs. A375 CTRL).

**Figure 5 antioxidants-12-00041-f005:**
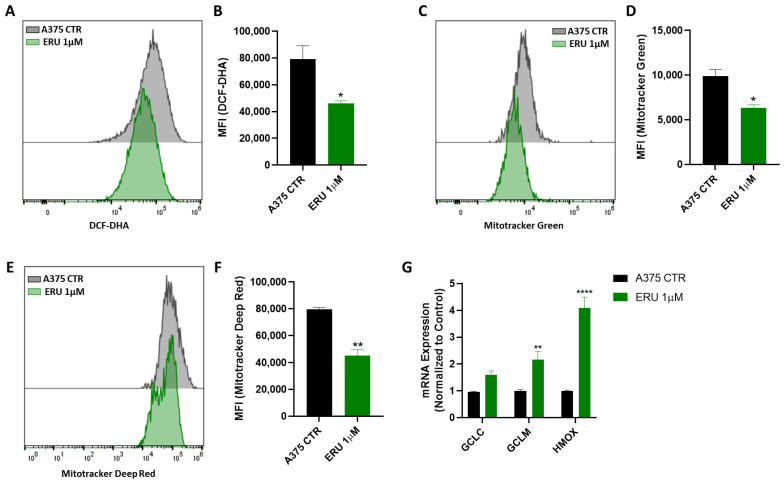
ERU inhibited ROS production in melanoma cells by limiting their mitochondrial function. (**A**–**F**) Representative examples of flow cytometry analysis of DCF-DHA. (**A**) MitoTracker Green (**C**) and MitoTracker Deep Red staining (**E**) in A375 cells that were untreated (black histograms) and those that underwent ERU treatment (green histograms) for 48 h with their respective quantification in terms of mean fluorescence intensity (MFI) (**B**,**D**,**F**). (**G**) Expression of GCLC, GCLM and HMOX assessed with qPCR analysis in A375 melanoma cells that were treated (green bar) or not treated (black bar) with 1 μM ERU. Data were shown as mean ± SEM of at least three independent experiments (* *p* < 0.05, ** *p* < 0.01 and **** *p* < 0.0001 vs. A375 CTRL).

## Data Availability

The data are contained within the manuscript.

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
