# Peer review of "Antiproliferative and Proapoptotic Effects of Erucin, a Diet-Derived H2S Donor, on Human Melanoma Cells"

_antioxidants, 2022, doi:10.3390/antiox12010041_

Round 1

Reviewer 1 Report

Daniela Claudia Maresca submitted a manuscript entitled ‚Anti-proliferative and proapoptotic effect of Erucin,
a diet-derived H2S donor, on human melanoma cells‘ in which they investigated the effect of erucin on human melanoma cells in vitro. Erucin is a diet derived component with supposedly anticancer properties due to its release of H2S. These anticancer effects were initially investigated on different melanoma cell lines, but only A375 melanoma cells were used for detailed analyses. Erucin was used in different concentrations derived from a dose response curve with five melanoma cell lines and one keratinocyte cell line. All of them showed significant reduction in proliferation rate when treated with erucin. The compont was shown to reduce proliferation, migration and clonogenic activity. Further apoptotisis induction was determined and cell cycle shift. An erucin induced change in melanin content was related to the migratory behavior and EMT transition as indicators for metastasis promotion. Lastly the effect on mitochondrial function was studied.

The manuscript is well and consicely written. The description of methods is clear and understandable. The experimental approach is very comprehensive and follows the intention of the research goal. The figures are represented in a clear colour scheme and are presented clearly.

There are only a few questions or remarks.

Why did the authors choose the authors this particula cell line? In figure 1 all tested cells showed a significant response towards erucin. What is special about A375 cells?

Why did the authors choose Hacat cells as comparison? These are a different cell type. As a comparison non-malignant melanocytes could be suitable. Please provide an explanation.

Regarding fig 4, I recommend to use a different term for ‚wound healing assay‘. It is correct that this assay is widely used as an in vitro wound healing assay but in this context it doesn‘t fit tot he question. The authors should refer to migration assay or similar in order to avoid confusion.

The authors used different concentrations of erucin for the different assay. Please explain a rationale for this choice.

For the most part, the discussion summarises the results. I would encourage the authors to add some current findings on similar diet derived compounds and relate their findings to currently available research. Other cancer cell types have been investiged, for example. How do your results relate to them?

Reviewer 2 Report

Having read the manuscript I have the following comments.

1.  What is the genetic makeup of A375 melanoma cells?  You do not mention this and why were BRAFWT melanoma cells not investigated in this study?

2.  HaCaT cells are keratinocytes in origin so why are they used as a negative control for melanoma cells?  why did you not use primary melanocytes?

3. L35-38 please note that the BRAFV600E mutation is not UV-induced, this statement is incorrect and needs to be rewritten.

4. L30 what is the third age you mention in "affects people belonging to the third age"?

5. The authors need to revise the manuscript and should assistance from a native English speaker to improve the grammatical errors that appear throughout the manuscript.

6.  The materials and methods section is poorly written and needs to be completely revised.  Units are missing or inconsistent, concentrations not listed in sections, times not stated, etc.  Align the primers so they are easy to read.  How did you trigger caspase activity in these cells?

7.  Why were different concentration of ERU used throughout this study, can you please explain your rationale for doing so?  

8.  Fig 1A is poorly drawn, the X axes are incorrect, and is not scientific in how it is presented.

9.  Why are the histograms all in grey and only the outside line coloured, please colour in the entire histogram.  It is difficult to see what each histogram is related to.

10.  You are correct in your interpretation of Fig 2C.  the top right hand sector represents both late apoptotic and necrotic cells, not just late apoptotic cells.  The top left sector represents cell debris not necrotic cells.  Please correct, your comments in the manuscript accordingly.

11.  Please include scale bars on Fg 4A & E.

12.  In many of the flow plots especially in Fg 5, what should be written on the Y axes?

13.  many statements in your discussion need to be referenced, which they are not, eg. L 376-7.

14.  You claim ERU releases H2S, yet nowhere in this manuscript do you show any evidence of this, why?

15.  Species names are always written in lower case so replace Savita on L13 with savita.

Reviewer 3 Report

The manuscript by Maresca and co-authors is very well written and data presentation is very well presented. Few minor issues need to be clarified, namely

1. in the abstract the species name (line 3) has both initials with capital letters, and the second should be revised to small letter

2. MTT assay is a viability evalution of the mithocondria activity not a proliferation assay, as CFSE! please amend

3. if the whole study focus on the A375 cells is it worth to evaluate the viability of WM1862, WM983A and WM983B cell lines? and if yes, why these cell lines?

Round 2

Reviewer 2 Report

I would like to thank the authors for addressing my concerns and comments in regards the initial version of this manuscript.  I strongly urge you in future to use melanocytes as the primary cell on which to compare melanoma cells, rather than using HaCaT cells.